

# Emergent relationships on burned area in global satellite observations and fire-enabled vegetation models

Matthias Forkel[1], Niels Andela[2], Sandy P. Harrison[3], Gitta Lasslop[4], Margreet van Marle[5], Emilio Chuvieco[6], Wouter Dorigo[1], Matthew Forrest[4], Stijn Hantson[7], Angelika Heil[8], Fang Li[9], Joe Melton[10], Stephen Sitch[11], Chao Yue[12], and Almut Arneth[13]

[1] Climate and Environmental Remote Sensing Group, Department of Geodesy and Geoinformation, Technische Universität Wien, Vienna, Austria
[2] Biospheric Sciences Laboratory, NASA Goddard Space Flight Center, Greenbelt, MD, USA
[3] Department of Geography and Environmental Science, University of Reading, Reading, United Kingdom
[4] Senckenberg Biodiversity and Climate Research Institute, Frankfurt am Main, Germany
[5] Deltares, Delft, The Netherlands
[6] Environmental Remote Sensing Research Group, Department of Geology, Geography and the Environment, Universidad de Alcalá, Alcalá de Henares, Spain
[7] Geospatial Data Solutions Center, University of California, Irvine, CA, USA.
[8] Department for Atmospheric Chemistry, Max Planck Institute for Chemistry, Mainz, Germany
[9] International Center for Climate and Environmental Sciences, Institute of Atmospheric Physics, Chinese Academy of Sciences, Beijing, China
[10] Climate Research Division, Environment Canada, Victoria, BC, Canada
[11] College of Life and Environmental Sciences, University of Exeter, Exeter, United Kingdom
[12] Laboratoire des Sciences du Climat et de l'Environnement, Gif-sur-Yvette, France
[13] Atmospheric Environmental Research, Institute of Meteorology and Climate research, Karlsruhe Institute of Technology, Garmisch-Partenkirchen, Germany

*Correspondence to*: Matthias Forkel (matthias.forkel@geo.tuwien.ac.at)

**Abstract.** Recent climate changes increases fire-prone weather conditions and likely affects fire occurrence, which might impact ecosystem functioning, biogeochemical cycles, and society. Prediction of how fire impacts may change in the future is difficult because of the complexity of the controls on fire occurrence and burned area. Here we aim to assess how process-based fire-enabled Dynamic Global Vegetation Models (DGVMs) represent relationships between controlling factors and burned area. We developed a pattern-oriented model evaluation approach using the random forest (RF) algorithm to identify emergent relationships between climate, vegetation, and socioeconomic predictor variables and burned area. We applied this approach to monthly burned area time series for the period 2005-2011 from satellite observations and from DGVMs from the Fire Model Inter-comparison Project (FireMIP) that were run using a common protocol and forcing datasets. The satellite-derived relationships indicate strong sensitivity to climate variables (e.g. maximum temperature, number of wet days), vegetation properties (e.g. vegetation type, previous-season plant productivity and leaf area, woody litter), and to socioeconomic variables (e.g. human population density). DGVMs broadly reproduce the relationships to climate variables





and some models to population density. Interestingly, satellite-derived responses show a strong increase of burned area with previous-season leaf area index and plant productivity in most fire-prone ecosystems which was largely underestimated by most DGVMs. Hence our pattern-oriented model evaluation approach allowed to diagnose that current fire-enabled DGVMs represent some controls on fire to a large extent but processes linking vegetation productivity and fire occurrence need to be

improved to accurately simulate the role of fire under global environmental change.

## 1 Introduction

About 3% of the global land area burns every year (Chuvieco et al., 2016; Giglio et al., 2013; Randerson et al., 2012). Fire represents a strong control on large-scale vegetation patterns and structure (Bond et al., 2004) and can significantly accelerate impacts of changing climate or land management on global ecosystems (Aragão et al., 2018; Beck et al., 2011). Fires affect

regional climate directly through changing surface albedo (Randerson et al., 2006), atmospheric trace gas and aerosol concentrations (Andreae and Merlet, 2001; Ward et al., 2012), and on longer time scales by affecting vegetation composition and structure with subsequent impacts on the carbon cycle and hydrology (Li and Lawrence, 2016; Pausas and Dantas, 2017; Tepley et al., 2018; Thonicke et al., 2001).

Climate influences several aspects of the fire regime, including the seasonal timing of lightning ignitions, temperature and

moisture controls on fuel drying, and wind-driven fire spread. Climate also influences the nature and availability of fuel, through its impact on vegetation productivity and structure (Harrison et al., 2010). Vegetation structure in turn influence the patterns of fuel amounts and moisture that directly determine fire spread, severity, and extent (Krawchuk and Moritz, 2011; Pausas and Ribeiro, 2013). People set and suppress fires and use them to manage agricultural and natural ecosystems, for land use change and deforestation practices (Andela and van der Werf, 2014; Marle et al., 2017). Human-induced modifications

and fragmentation of natural vegetation through agricultural expansion and urbanization limits fire spread (Bowman et al., 2011). Thus, climate, vegetation, and human controls on fire are multivariate and have strong interactions with one another (Bowman et al., 2009; Harrison et al., 2010; Krawchuk et al., 2009). Empirical analyses of fire regimes by using machine learning algorithms have identified the most important variables and their sensitivities for fire occurrence and spread (Aldersley et al., 2011; Archibald et al., 2009; Bistinas et al., 2014; Forkel et al., 2017; Krawchuk et al., 2009; Moritz et al.,

2012). However, because of the difficulty of factoring out interactions between predictor variables, such sensitivities represent emergent relationships rather than specific physical controls on fire. Thus, it has proved difficult to disentangle the role of changes in any single factor on the trajectory of changes in fire regimes. For example, changes in climate result in increasing fire weather conditions and fire activity in some temperate regions (Holden et al., 2018; Jolly et al., 2015; Müller et al., 2015) but it has been suggested that changes in land use compensate climate effects and result for example in declining burned areas

in African Savannahs (Andela and van der Werf, 2014). Hence there is still uncertainty, for example, about the cause of the recent observed decline in global burned area (Andela et al., 2017). There is even greater uncertainty about the potential trajectory of changes in fire regimes in the future (Settele et al., 2014).




Fire-enabled Dynamic Global Vegetation Models (DGVMs) or Earth System Models are process-oriented tools to predict the consequences of future climate change on fire regimes and associated feedbacks (Hantson et al., 2016). Our faith in these projections is contingent on the ability of these models to capture features of the current situation. State-of-the-art fire-enabled DGVMs partly capture the spatial patterns of burned area (Andela et al., 2017; Kelley et al., 2013) but doubt has been cast on

their ability to capture the response to extreme events and recent trends in burned area (Andela et al., 2017). This suggests that these models inaccurately represent the response of fire to combined changes in climate, vegetation, and socioeconomic drivers.

Here we aim to test how fire-enabled DGVMs reproduce emergent relationships with the drivers of fire activity. We apply a machine learning algorithm to the output from seven fire-enabled DGVMs and a suite of satellite and other observation-based

datasets in order to derive emergent relationships between a number of potential drivers of fire activity and burned area. By comparing the model- and data-derived emergent relationships, we assess the degree to which DGVMs reproduce these relationships. While we make no assumption about the actual physical controls on burned area, this comparison allows us to pinpoint relationships between drivers and burned area that are unrealistic represented in fire-enabled DGVMs.

## 2 Data and Methods

### 2.1 Method summary

In order to infer relationships between potential drivers on fire in satellite data and fire-enabled DGVMs, we applied the random forest (RF) machine-learning algorithm to predict monthly burned area (response variable) from climate, vegetation, and socioeconomic predictor variables (Figure 1). Predictor variables and burned area were taken either from satellite and other observation-based datasets or from simulations by a suite of fire-enabled DGVMs from the Fire Model Inter-comparison

project (FireMIP) (Rabin et al., 2017) to derive relationships for datasets and models, respectively.

The RF algorithm is a regression approach that allows non-linear, non-monotonic, and non-additive relations between multiple predictor variables and the target variable. RF averages predicted values across an ensemble of decision trees that are built based on the training data set (Breiman, 2001; Cutler et al., 2012). We built three sets of RF models. We first built RF models for satellite-observed burned area based on a multitude of predictor variables to derive relationships from data. There are

differences between the available burned area datasets; we therefore used five recent and/or well-established datasets to encompass these uncertainties. The fire-enabled DGVMs do not use some of the predictor variables in the satellite-derived RF. We hence built a second set of satellite-derived RF models with a reduced set of predictor variables. The third set of RF models was derived for each FireMIP model by using the simulated burned area as target variable and simulations of gross primary production (averaged over precedent months), biomass and land cover predictor variables, and the population density

and climate predictor variables that were used as inputs for the models (according to the FireMIP protocol).

From each RF model we then derived the importance (sect. 2.7), relationships, and sensitivity (sect. 2.8) of each predictor variable to burned area. Relationships and sensitivities were derived by computing individual conditional expectation (ICE)





and partial dependencies curves (Goldstein et al., 2013). These dependencies represent the emergent relationships of burned area to drivers in the observation- or model-variable space. We then compared the data- and model-derived emergent relationships and sensitivities both globally and per grid cell basis.

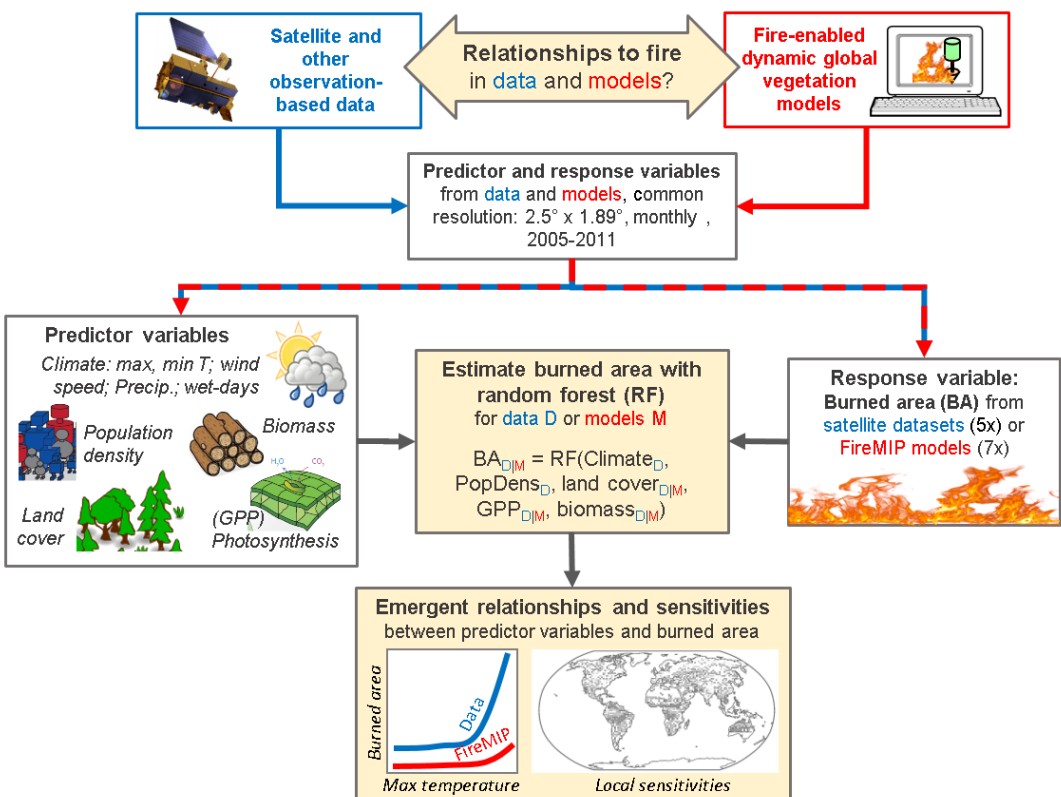

**Figure 1: Overview of the approach on using the random forest machine learning algorithm to derive emergent relationships between several predictor variables for burned area in satellite and other observation-based data and in fire-enabled DGVMs.**

## 2.2 Burned area from satellite datasets

There are several global burned area datasets, and both the spatial patterns and temporal dynamics differ between them (Hantson et al., 2016; Humber et al., 2018) because they use different satellite sensors and retrieval algorithms, and have different sensitivities to small fires (Chuvieco et al., 2016; Giglio et al., 2013; Randerson et al., 2012). We used the variability between five global datasets (Table 1) as an estimate of uncertainty. However, by doing so we might still underestimate the real uncertainty in burned area observations because all datasets rely on active fire detections (thermal anomaly) and on





reflectance changes from the same sensor (MODIS). As exception, CCI_MERIS uses MERIS reflectances combined with MODIS active fires.

We restricted our analysis to burned area data with high observational quality. Observational quality indicates to which degree missing input satellite imagery or contaminations by clouds, smoke, snow and shadows limit burned area detection. Especially

MERIS land observations are subject to substantial gaps in raw data acquisitions (Tum et al., 2016). Low observational coverage can result in strongly underestimated burned area. Here, we used the CCI_MERIS "observed area fraction" layer as a time-variant mask to all burned area datasets and only included estimates for months with observational coverage higher than 80 %. We also excluded burned area in months with < 0 °C to remove suspicious small burned areas in polar regions or in winter months that are likely caused by insufficiently corrected gas flares and other industrial activities. Analyses were made

with monthly burned area observations for the period 2005-2011, which is the common period between the five datasets.

## 2.3 Burned area from FireMIP models

A detailed description of FireMIP DGVMs and the simulation protocol is given by Rabin et al. (2017). Here we used monthly burned area from seven models that made transient simulations from 1700 to 2013 (Table 1, bottom half). The models were forced using inputs of meteorological variables from the CRUNCEP V5 dataset (Wei et al., 2014), monthly cloud-to-ground

lightning strikes (Rabin et al., 2017), annually-updated values of human population density from the HYDE 3.1 data set (Klein Goldewijk et al., 2010), annually-updated land use and land cover changes from the Hurtt et al. (2011) data set, and annually-updated values of global atmospheric CO2 (Le Quéré et al., 2014). Although forcing datasets are common across DGVMs, they do not use the same set of forcing variables, i.e. wind speed (WSPEED), or use population density (PopDens) for fire ignitions and/or fire suppression.

The model outputs were aggregated to a common spatial resolution of 2.5° longitude x 1.89° latitude. Analyses were made for the same period as the common window of the satellite data (2005-2011) and by also applying the "observed area mask" from the satellite data.



**Table 1: Overview of used burned area datasets and FireMIP models.**

| Abbreviation used in this study | Satellite dataset or FireMIP model | Spatial resolution and temporal coverage | Satellite sensor (all datasets use thermal anomalies from MODIS) or model characteristics | Reference |
|---|---|---|---|---|
| **Satellite-derived burned area datasets** | | | | |
| GFED4 | GFED4 | 0.25° x 0.25° 1995-2015 | Based on MODIS collection 5 (500 m) (Giglio et al., 2009) | (Giglio et al., 2013) |
| GFED4s | GFED4s | 0.25° x 0.25° 1995-2015 | Based on GFED4 with additional estimation of small fires | (Randerson et al., 2012) |
| CCI_MERIS | ESA Fire_cci V4.1 | 300 m 2005-2011 | MERIS V4.1 reflectances | (Chuvieco et al., 2016) |
| CCI_MODIS | ESA Fire_cci V5.0 | 250 m 2000-2015 | MODIS V5.0 | (Chuvieco et al., 2018) |
| MCD64C6 | MCD64C6 | 500 m 2000-2018 | MODIS collection 6 | (Giglio et al., 2018) |
| **FireMIP models** | | | | |
| CLM | CLM Li et al. fire module | 2.5° x 1.89° 1700-2013 | Uses WSPEED for fire spread Uses PopDens for ignitions and suppression | (Li et al., 2012, 2013) |
| CTEM | CTEM | 2.8125° x 2.8125° 1700-2013 | Uses WSPEED for fire spread Uses PopDens for ignitions and suppression | (Arora and Boer, 2005; Melton and Arora, 2016) |
| JSBACH | JSBACH-SPITFIRE | 1.875° x 1.875° 1700-2013 | Uses WSPEED for fire spread Uses PopDens for ignitions and suppression | (Lasslop et al., 2014) |
| JULES | JULES-Inferno | 1.25° x 1.875° 1700-2013 | Empirical model No WSPEED Uses PopDens for ignitions only | (Mangeon et al., 2016) |
| LPJG-SIMF | LPJ-GUESS-SIMFIRE-BLAZE | 0.5° x 0.5° 1700-2013 | Empirical model with seasonal dynamic from GFED3 dataset No WSPEED for fire spread Uses PopDens for fire suppression | (Knorr et al., 2014, 2016) |
| LPJG-SPITF | LPJ-GUESS-SPITFIRE | 0.5° x 0.5° 1700-2013 | Uses WSPEED for fire spread Uses PopDens for ignitions | (Lehsten et al., 2010, 2016) |
| ORCHIDEE | ORCHIDEE-SPITFIRE | 0.5° x 0.5° 1700-2013 | Uses WSPEED for fire spread Uses PopDens for ignitions | (Yue et al., 2014, 2015) |

**2.4 Evaluation of data-data and model-data temporal agreement**

We evaluated the temporal agreement of monthly burned area time series in 2005-2011 between the datasets and between the

5    datasets and the fire-enabled DGVMs based on various model performance metrics (Janssen and Heuberger, 1995) on a per-



grid cell basis. We selected the Spearman rank-correlation coefficient to compare the temporal agreement and the fractional variance (FV) to compare the variability of burned area per grid cells:

$$FV = \frac{\sigma_x - \sigma_{ref}}{0.5 \times (\sigma_x + \sigma_{ref})} \tag{1}$$

where $\sigma_{ref}$ and $\sigma_x$ are the variances of the reference and observed or simulated burned area, respectively. FV ranges between -

2 and 2 where negative values indicate an underestimation and positive values an overestimation of the observed variance. The reference *ref* is a vector of monthly burned area time series from all satellite datasets:

$$ref = [BA.CCI_{MERIS}, BA.CCI_{MODIS}, BA.GFED4, BA.GFED4s, BA.MCD64C6]$$

In the case of a comparison of a single satellite datasets (e.g. x = BA.CCI_MERIS) with the other satellite datasets, this dataset was not used in the reference vector. This approach directly considers the differences between datasets in the computation of

model performance metrics and implies that it is impossible for a FireMIP model or for one single satellite dataset to reach an optimal correlation of unity or a FV of zero as long as the satellite burned area datasets show differences. We used the median of the correlation coefficient and of the FV for each grid cell to quantify the data-data or model-data agreement over the ensemble of datasets or models. As a single global agreement metric, we computed the percentage of the land area that shows a "good" agreement from the spatial patterns of Spearman correlation and FV, where good agreement for an individual grid

cell was defined as having a correlation $\geq 0.25$ and $-0.75 \leq FV \leq 0.75$.

## 2.5 Predictor variables and datasets

Several variables have been identified as predictors of global fire in previous studies, *inter alia* the number of dry or wet days per month (WET), diurnal temperature range (DTR), maximum temperature (TMAX), grass and shrub cover, leaf area index (LAI), net primary production (NPP), population density (PopDens), and gross domestic product (GDP) (Aldersley et al.,

2011; Bistinas et al., 2014). Other variables have been found important for fire at a regional scale, including total precipitation, tree cover, forest cover type, tree height, biomass and litter fuel loads, and grazing (Archibald et al., 2009; Chuvieco et al., 2014; Parisien et al., 2010; Pettinari and Chuvieco, 2017). We created a combined set of potential variables used in these studies to predict burned area (Table A 1). We used data on gross primary production (GPP) instead of NPP as GPP can be estimated from eddy covariance observations and does not require model assumptions about autotrophic respiration.

**Climate data.** Climate data was taken from the CRUNCEP V5 dataset (Wei et al., 2014). CRUNCEP provides six-hourly time series of precipitation, maximum and minimum temperature, and wind speed. From these time series, we derived the monthly mean of daily maximum temperature (CRUNCEP.TMAX) and minimum temperature (CRUNCEP.TMIN), the monthly mean daily diurnal temperature range (CRUNCEP.DTR = TMAX – TMIN), the monthly 90th percentile of daily wind speed (CRUNCEP.WSPEED), monthly total precipitation (CRUNCEP.P) and the number of wet days per month (CRUNCEP.WET).

A wet day was defined as a day with $\geq 0.1$ mm precipitation (Harris et al., 2014).

**Land cover.** Land cover was taken from the ESA CCI Land cover V2.0.7 dataset which provides annual land cover maps for the period 1992-2015 (Li et al., 2018). Land cover classes were converted into the fractional coverage of plant functional types



(PFTs). For this conversion, we used the cross-walking approach (Poulter et al., 2011, 2015) based on the conversion table in Forkel et al. (2017). Individual PFTs combine growth form (tree, shrubs, herbaceous vegetation, or crops) with leaf type (broad-leaved or needle-leaved) and leaf longevity (evergreen or deciduous). The variable Tree.BD, for example, is the fractional coverage of broad-leaved deciduous trees (Table A 1). We created an additional category combining trees and shrubs

(e.g. TreeS.BD = Tree.BD + Shrub.BD) because most of the FireMIP models simulate woody vegetation rather than separating shrubs and trees explicitly (Table S 1). JULES, LPJG-SIMF, and LPJG-SPITF dynamically simulate the fractional coverage of PFTs, but CLM, CTEM, JSBACH, and ORCHIDEE used prescribed PFT distributions. We reclassified the PFTs of each model into the same set of PFTs that we derived from the CCI land cover dataset (Table S 1).

**Vegetation productivity.** Data on gross primary production (GPP) and leaf area index (LAI) were taken to account for the

seasonal effects of vegetation productivity and canopy development. GPP was taken from the FLUXCOM dataset which is up-scaled from GPP estimates at FLUXNET measurement sites (Tramontana et al., 2016). We used the FLUXCOM dataset that used satellite and CRUNCEP meteorological data for the upscaling. LAI was taken from MODIS (USGS, 2001, p.2). GPP and LAI were averaged to monthly mean values (e.g. variable name GPP.orig). To account for seasonal fuel accumulation, we also computed previous-season GPP or LAI values as the mean over the three and six months before the month of comparison

with burned area (e.g. GPP.pre3mon and GPP.pre6mon).

**Biomass and fuels.** We used temporally-static vegetation datasets to account for the effects of vegetation biomass, fuel properties, and ecosystem structure on burned area dynamics. Total above- and below-ground vegetation biomass was obtained from Carvalhais et al. (2014), which is based on an above-ground forest biomass map for the tropics for the early 2000s (Saatchi et al., 2011), a total forest biomass map for temperate and boreal forests for the year 2010 (Thurner et al., 2014), and

an estimate of herbaceous biomass (Carvalhais et al., 2014). From each FireMIP model, we used the simulated vegetation carbon averaged for the years 2005-2011 as the equivalent to this data set. We used canopy height from Simard et al. (2011); this data set provides a snapshot of average canopy height in 2005. Factors related to fuel properties, specifically grass height, litter depth, woody litter depth, and amounts of woody litter in different size classes were extracted from the global fuelbed database (Pettinari and Chuvieco, 2016). This database is based on a land cover-based extrapolation of regional fuel databases

to the globe and provides a generic picture of conditions around 2005.

**Socioeconomic data.** We used the annually-varying population density dataset from the HYDE V3.1 database (Klein Goldewijk et al., 2011), which was used as a forcing dataset for the FireMIP simulations. We also used annually-varying gross domestic product per capita (GDP) (World Bank, 2018), a static satellite-derived index of socio-economic development based on night-time lights for the year 2006 (Elvidge et al., 2012), and a dataset on cattle density for the year 2007 (Wint and

Robinson, 2007).

## 2.6 Random forest experiments and selection of predictor variables

We performed our analysis using the randomForest package V4.6-12 in R (Liaw and Wiener, 2002). We trained the RF with 500 regression trees. The training target was either a "satellite-observed" or a "model-simulated" burned area, i.e. we trained



one RF against each burned area dataset and each individual FireMIP model simulation, respectively. We used two sets of predictor variables in three sets of RF experiments (Table A 1):

- "RF.Satellite.full" for satellite-derived RF experiments: We used 23 out of all 28 predictor variables to train RF models for each burned area dataset. Five predictor variables were not included in the RF because they were highly correlated with others (r > 0.8, i.e. night-light development index, cattle density, woody litter for the 10 h fuel size class, precedent 3-monthly GPP, and precedent 3-monthly LAI, Figure S 1). The purpose of these experiments was to identify the relationship between burned area and each predictor variable from datasets.

- "RF.Satellite.fm" for satellite-derived RF experiments: These experiments were also trained against burned area datasets but included only the reduced set of 16 data-based predictor variables that are available from both observational datasets and the FireMIP (fm) models.

- "RF.FireMIP.fm" for model-derived RF experiments: These experiments used the reduced set of predictor variables with land cover, GPP, biomass, and the response variable burned area taken from simulations of each FireMIP model. The purpose of these experiments was to compare relationships and sensitivities from satellite- and FireMIP-derived RF experiments.

**2.7 Importance of predictor variables in random forest**

The normal method of determining the importance of predictor variables for RFs (increment in mean-squared error, MSE) was found to be overly sensitive to the burnt area dataset that was used in training because of the highly skewed distribution of burned area, and this hampers its interpretability (Figure S 8, Figure S 9). To overcome this issue and to obtain additional information about regional (i.e. grid cell-level) importance of predictor variables, we developed an alternative approach. This alternative approach uses the fractional variance (FV) and Spearman correlation (r) instead of the MSE and is computed for each grid cell. The importance of variables is quantified as a distance D in a two-dimensional space based on these metrics:

$$D = \sqrt{\left(0.5 \times (FV_p - FV_0)\right)^2 + \left(r_p - r_0\right)^2} \qquad (2)$$

where $FV_0$ and $r_0$, and $FV_p$ and $r_p$ are the performance metrics based on the original RF predictions and based on the RF predictions after permuting a single predictor variable, respectively. The FV-related term was multiplied with 0.5 to obtain the same range like the correlation. FV and r are computed at grid cell-level based on monthly burned area time series from the RF predictions and the training data (i.e. burned area from a satellite dataset or from a FireMIP model). As the metric D depends on the permutation, we permutated each predictor variable 10 times and averaged the D metric.

**2.8 Deriving emergent relationships and sensitivities from random forest**

Insight into the shape of a relationship between a predictor and the target variable in a trained RF can be obtained from partial dependence (PDP) (Friedman, 2001) and individual conditional expectation (ICE) plots (Goldstein et al., 2013) (Figure S 2). PDPs show the partial relationship between the predicted target variable and one predictor variable when other predictor



variables are set to their mean value. ICE plots show the relationship between the predicted target variable and one predictor variable for individual cases of the predictor dataset (Goldstein et al., 2013). In our application, an individual case is a specific combination of climate, land cover, vegetation, and socioeconomic data for a given grid cell in a given month (Figure S 2). The average of all ICE curves corresponds to the PDP. We used the ICEbox package V1.1.2 for R for the computation of ICE

curves and partial dependencies (Goldstein et al., 2013).

We computed ICE curves for all predictor variables and from all RF experiments (Supplementary Information 4 and 5). We computed ICE curves and PDPs based on the global dataset to analyse and compare global emergent relationships. Pearson's correlation coefficient was computed between pairs of satellite- and model-derived ICE curves to quantify the agreement of the emergent relationships (Figure S 15). We also computed PDPs for each grid cell to produce global maps of partial

sensitivities for selected predictor variables. To summarize and map the PDP of each grid cell in a single number, we fitted a linear quantile regression to the median between the partial dependence of burned area and the corresponding predictor variable and mapped the slope of this regression. In the following, we name this slope "sensitivity".

## 3 Results

### 3.1 Evaluation of temporal burned area dynamics

Here we compare the monthly temporal dynamics of burned area from the satellite datasets, FireMIP model simulations, and random forest predictions for the overlapping period 2005-2011. The satellite datasets showed in average relatively good agreement with each other (i.e. "good" is $r \geq 0.25$ and $-0.7 \leq FV \leq 0.7$) over 70% of the global vegetated land area, with best agreement in frequently burning grasslands and savannahs (Figure 2 a). However, individual datasets showed good agreement in only 31-56% of the land area (Figure S 3). The largest dissimilarities between burned area datasets occurred in temperate

land use-intense regions (North America, Europe, China), tropical forests, and in sparsely vegetated arid and tundra regions. These difference are likely caused by limited detection possibilities under cloud cover (e.g. in the Amazon) and by the sensitivities of the algorithms to detect small fires (temperate and sparsely-vegetated regions). As the CCI_MERIS dataset is based on a different sensor, it is the most different from the other datasets (31% of land area with good agreement, Figure S 3). Hence these uncertainties make it necessary to train RF to each dataset separately in order to assess how such uncertainties

translate into emergent relationships to burned area.

FireMIP models showed good agreement with satellite datasets in 9% of the land area (Figure 2 b). In particular, models tended to underestimate the variability of burned area in key biomass burning regions, while overestimating fire variability in arid and some temperate regions of infrequent fire activity. Individual FireMIP models had weaker performance than the model ensemble (6% to 8% with good agreement, Figure S 4).





**Figure 2: Comparison of temporal burned area dynamics from satellite datasets, fire-enabled DGVMs, and random forest. The maps show the median Spearman rank-correlation coefficient and median fractional variance of the monthly burned area in 2005-2011 between (a) satellite datasets and the other satellite datasets (Figure S 3), (b) FireMIP model simulations and all satellite datasets (Figure S 4), (c) predicted burned area from RF and all satellite datasets ("full" set of predictor variables, Figure S 5), and (d) predicted burned area from RF trained against FireMIP models and the corresponding simulated burned area from each FireMIP model (Figure S 7). Green percentage numbers indicate land area with "good" agreement (Correlation ≥ 0.25 and -0.7 ≤ FV ≤ 0.7).**

The RF models can reproduce the temporal dynamics of the satellite burned area datasets reasonably well in most frequently

10    burning regions (Figure 2 c). The overall proportion of the vegetated land area showing good agreement in "full" experiments



was only 36% but individual RF models reached better performances (up to 41% with good agreement, Figure S 5). The "fm" RF models had slightly weaker performance (22% to 38% with good agreement, Figure S 6). However, the performance of RF models was much higher than the performance of FireMIP models (Figure 2 b). RF was also able to largely emulate the simulated burned area from FireMIP models (85% with good agreement with the FireMIP simulation, Figure 2 d). The RF

models most closely emulated simulated burned area in those FireMIP models that are based on empirical relationships (JULES, LPJG_SIMF, Figure S 7). In summary, the ability of RF models to emulate simulated or observed monthly burned area dynamics is sufficient for the purposes of comparing satellite-derived and FireMIP-derived relationships.

**3.2 Importance of predictor variables in random forest**

Satellite-derived RF experiments show that temperature-related variables were the most important predictors for temporal

burned area dynamics in temperate and boreal regions, and land cover- or productivity-related variables were most important in subtropical and tropical regions (Figure 3 a). Maximum temperature had on average the highest importance globally and was the most important predictor in 30-40% of the land area in satellite-derived RF (Figure 3 c and e). Productivity and land cover-related variables (i.e. mostly precedent 6-monthly GPP and broad-leaved deciduous tree cover in Savannahs) were the most important predictors in another 20-30% of the land area. Dryness-related predictor variables (WET and P) were most

important in tropical forest regions. Human-related predictor variables were only most important in a few grid cells, whereby cropland cover was, on average, of higher importance than population density (Figure 3 c). The satellite-derived importance was very similar among the burned area datasets (Figure 3 e).

On average, the FireMIP model-derived RF experiments broadly reproduced the satellite-derived importance of predictor variables (Figure 3 b). However, maximum temperature, precedent 6-monthly GPP, and number of wet days had a lower

importance, but diurnal temperature range, cropland cover, and precipitation had a higher importance than in the satellite-derived RFs (Figure 3 d). In addition, the model-derived importance of predictor variables differed among FireMIP models (Figure 3 e, Figure S 10). Most model-derived RF experiments underestimated the importance of precedent 6-monthly GPP and showed large differences in the importance of land cover-related predictors. The strongly varying size of the yellow and green bars in Figure 3 e indicate that differences in simulated burned area between FireMIP models mostly originate from how

productivity and land cover effects on fire are represented.





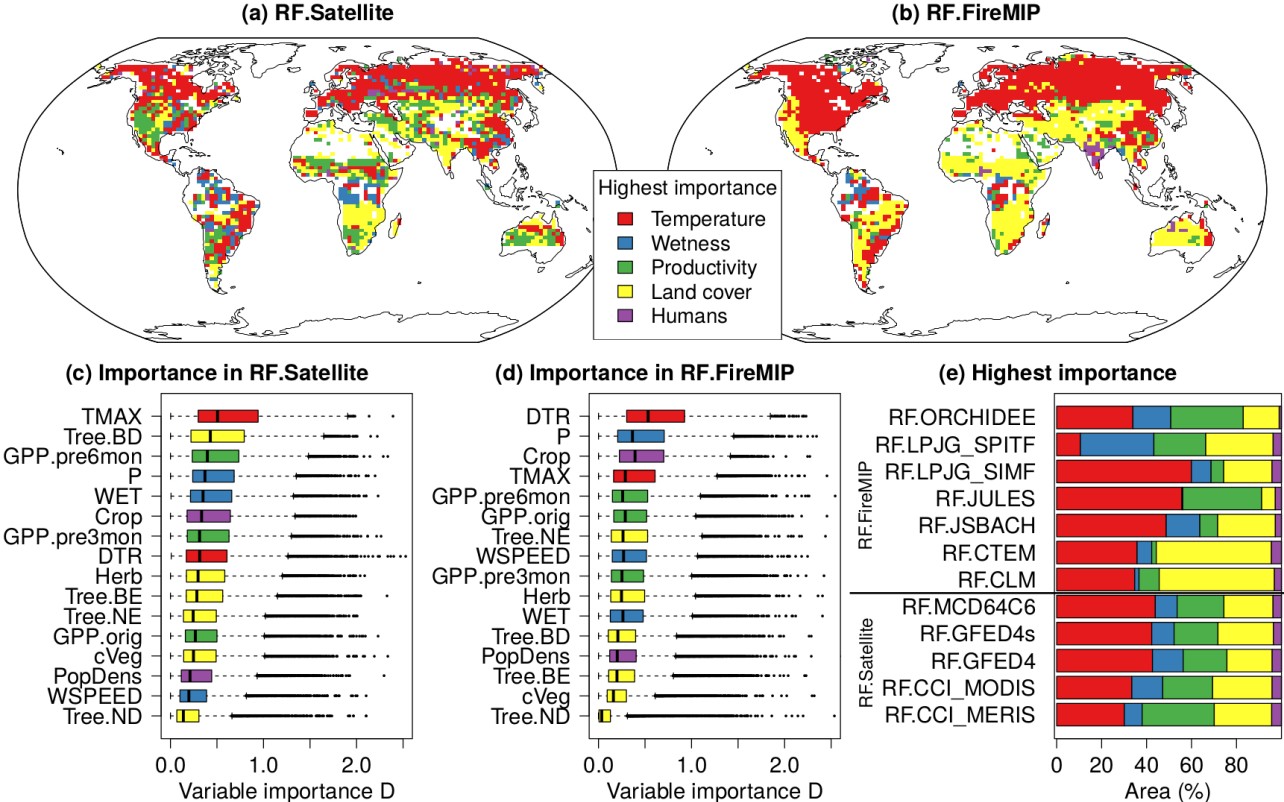

**Figure 3: Grid cell-level importance of predictor variables in satellite- and FireMIP-derived RF experiments.** Importance of variables is quantified as the change in the grid-cell level performance of the RF predictions after a predictor variable is permuted (D metric, see Methods). (a and b) Maps of the group of variables with the highest importance. For example, "temperature" (red) indicates that either TMAX or DTR had the maximum D metric and highest importance in a grid cell. (c and d) Global distributions of D for each variable from satellite- and model-derived "fm" RF experiments, respectively. Variables with the same colour are grouped together for the figures in panels (a, b, and e). (e) Area distribution of the variable groups with the highest D for each RF experiment.





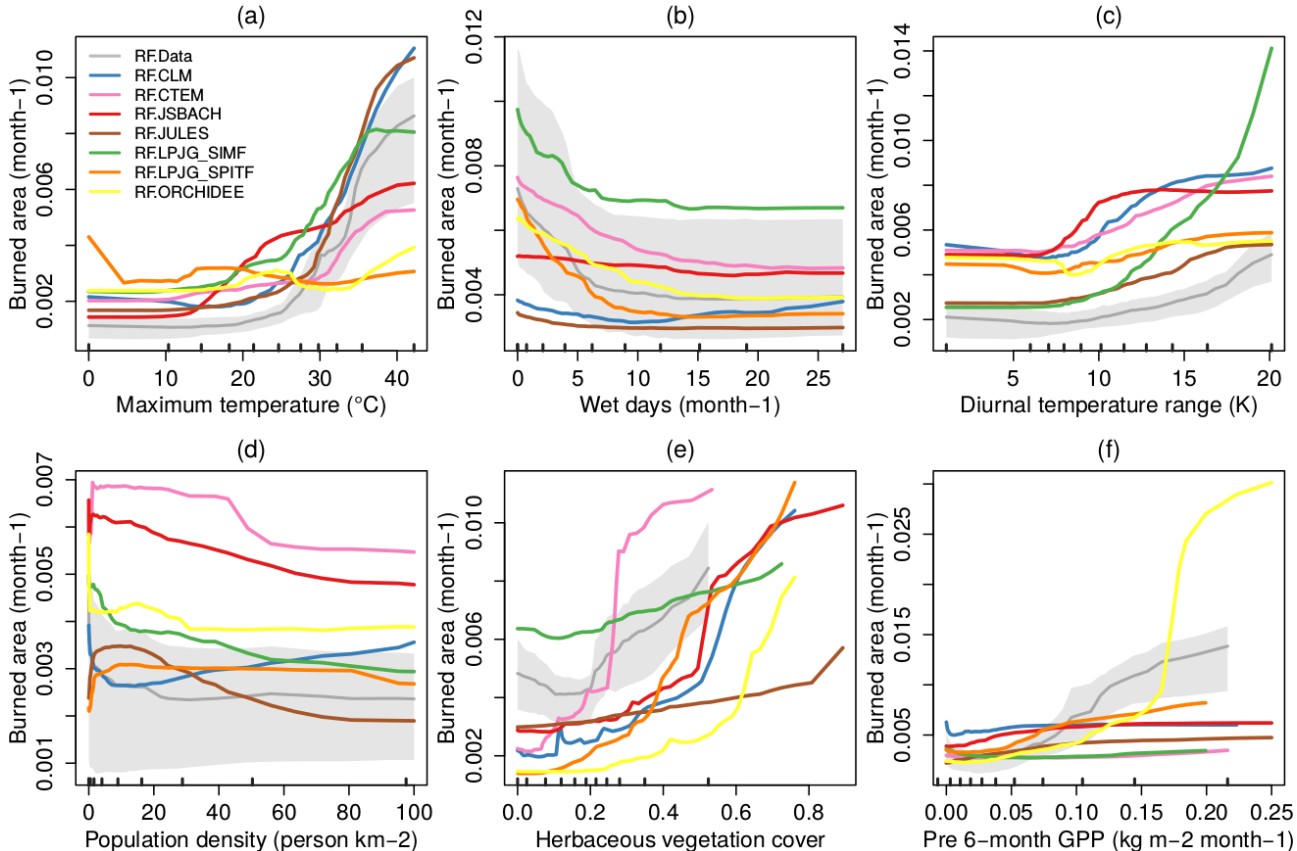

**Figure 4: Example of global emergent relationships of the fractional burned area per month to predictors from satellite-derived (grey, mean and range based on the five burned area datasets) and FireMIP model-derived (colours) "fm" random forest experiments for six selected variables. Tick marks along the x-axis show the deciles (minimum, quantile 0.1 to maximum) of the**
5 **global distribution of each predictor variable. Maximum temperature was cut at 0°C in (a) and population density was cut at 100 person km-2 in (d). Emergent relationships for other predictor variables are shown in Figure S 16 and Figure S 17.**

### 3.3 Emergent relationships of burned area to driving factors

#### 3.3.1 Climate

The satellite-derived global relationships showed expected patterns between burned area and climate variables: burned area increased exponentially with maximum temperature, decreased with an increasing number of wet days per month, and increased with diurnal temperature range (Figure 4 a-c). The shapes of the relationships of burned area to climate variables were robust among the burned area datasets (Figure S 11). However, burned area datasets show offsets between the relationship curves: For example, the curves that were derived from the GFED4s and CCI_MERIS datasets show usually higher burned

area than the curves from the other datasets (Figure S 11). These positive offsets are caused by the fact that GFED4s and



CCI_MERIS include more small fires and have hence an overall higher burned area than the other datasets. RF experiments that either use the "full" or "fm" set of predictor variables resulted in largely similar relationships (Figure S 12).

The relationships between burned area and climate variables were broadly similar for the FireMIP models (Figure 4 a-c, Figure S 15, Figure S 16). Most model-derived global relationships agreed relatively well (r > 0.5) with satellite-derived relationships

for maximum temperature, diurnal temperature range, and the number of wet days (Figure 5). However, LPJG-SPITF and ORCHIDEE did not reproduce the satellite-derived increase of burned area with maximum temperature (Figure 4 a). In the case of LPJG_SPITF, this is likely due to a modification to the calculation of dead fuel moisture. In contrast to other SPITFIRE implementations, LPJG_SPITF uses soil moisture in part to determine dead fuel moisture. This likely explains the failure of LPJG_SPITF to reproduce the dependency on maximum temperature and the markedly different behaviour from the other

SPITFIRE models seen here. CLM and JSBACH did not reproduce the decrease of burned area with increasing number of wet days (Figure 4 b).

Regionally, sensitivities to maximum temperature were positive over most land areas in satellite-derived RF experiments (Figure 6 a, Figure S 18). Regional sensitivities to the number of wet days were negative in most land areas but were positive in arid regions and in boreal regions of Northern America (Figure 6 d, Figure S 20). Most FireMIP models tended to

overestimate the regional sensitivities between maximum temperature and burned area in comparison to the satellite-derived sensitivities in most non-forested regions (Figure 6 b-c). Regional sensitivities to wet days were very different among FireMIP models and in comparison to the satellite-derived sensitivities (Figure 6 e-f, Figure S 21). In summary these results show that fire-enabled DGVMs broadly reproduced the overall relationships and sensitivities of burned area with climate variables.

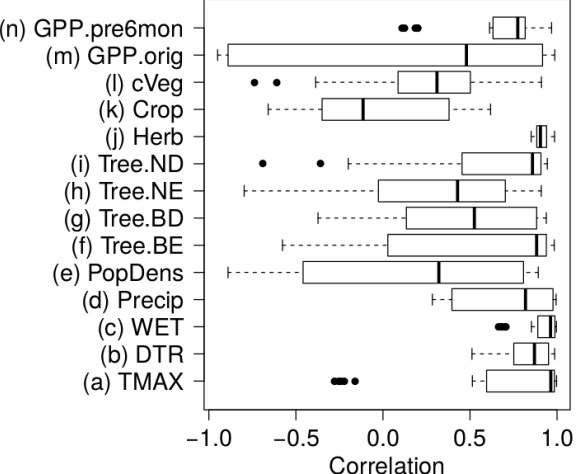

**Figure 5: Correlations between global relationships from satellite-derived and model-derived RF "fm" experiments. Pearson correlations were computed from the relationships as shown in Figure 4. Boxes show the distribution of all model-data correlations (5 satellite-derived relationships x 7 FireMIP model-derived relationships). Correlations for individual satellite- and model-derived RFs are shown in Figure S 15.**







Figure 6: Regional sensitivities of burned area to the driving factors for six selected variables (a-c) maximum temperature, (d-f) number of wet days per month, (g-i) population density, (j-l) herbaceous vegetation cover, and (m-o) precedent 6-monthly GPP. Sensitivities are slopes of a linear quantile regression fit to grid cell-level partial dependencies between burned area and the predictor variables as derived from satellite-derived "fm" RF experiments (left panel) and model-derived RF experiments (middle panel). The



**right panel shows the difference between model- and satellite-derived sensitivities. Stippling indicates locations where fewer than two model-derived sensitivities are within the range of satellite-derived sensitivities. The missing regions in southern Australia and New Zealand are due to missing data in the used vegetation carbon data set in satellite-derived RF experiments. Sensitivities for individual satellite datasets and FireMIP models are shown in Figure S 18 to Figure S 27.**

### 3.3.2 Socioeconomics

The satellite-derived global relationships showed that burned area increased exponentially as population density decreased at very low values (< 20 persons $km^{-2}$) and, generally, showed no sensitivity when population density was > 40 persons $km^{-2}$ (Figure 4 d, Figure S 13 a). Regionally, the satellite-derived sensitivity to population density varied with vegetation type. It was negative in most grassland and savannah ecosystems but positive in infrequently burning forested ecosystems (Figure 6 g). Burned area exponentially increased at very low gross domestic product per capita (Figure S 13 b). The relationship between burned area and cropland area was non-monotonic: all datasets showed a burned area peak at < 5% cropland, minimum burned area at 5-30% cropland cover, and an increasing burned area at > 30% cropland cover (Figure S 13 c). The satellite-derived relationships with cropland cover had only moderate correlations with the other satellite-derived relationships for some datasets (e.g. r = 0.53 for CCI_MODIS, Figure S 15 k) because global burned area products are not very accurate for agricultural fires (Hall et al., 2016).

The relationships between burned area and population density were very different among FireMIP models and partly in comparison to the satellite-derived relationships (Figure 4 d, Figure S 23). ORCHIDEE, LPJG_SIMF, and partly CLM and JSBACH reproduced the satellite-derived decline of burned area with increasing population density (r > 0.4, Figure S 15). LPJG_SPITF, CTEM, and JULES had a weak agreement with the satellite-derived sensitivities (r < -0.34). However, the model ensemble median reproduced the regionally negative relationships in savannahs and the partly positive relationships in forest regions (Figure 6 h-i). FireMIP model sensitivities to cropland cover showed large differences in comparison to satellite-derived sensitivities (Figure S 16 g). Only LPJG_SIMF reached a comparable correlation (r = 0.41) to the satellite-derived sensitivity because its internal formulation reduces burned area with increasing cropland cover, it however does not simulate crop fires. These large differences in the sensitivities of burned area to socioeconomic variables demonstrate that fire-enabled DGVMs mostly disagree on how human effects on fire should be represented.

### 3.3.3 Land cover, vegetation productivity, and biomass

The satellite-derived global relationships to vegetation-related predictor variables showed that burned area increased with increasing herbaceous vegetation cover (Figure 4e), with precedent 6-monthly GPP (Figure 4 f), with precedent 6-monthly LAI (Figure S 14 b), and with woody litter (Figure S 14 h). The satellite-derived relationships were for most land cover types and for vegetation carbon moderately to highly correlated (Figure S 15 f-o). Regionally, the satellite-derived relationship to herbaceous vegetation cover was positive in most ecosystems but negative in agricultural areas in Europe, India, Eastern Asia, and North America (Figure 6 j). The regional sensitivity to precedent 6-monthly GPP was strongly positive in most semi-arid





regions (Figure 6 m). These relationships reflect the importance of plant productivity and fuel production for burned area. Burned area decreased with increasing actual-month LAI (Figure S 14 a-d), reflecting the fact that fires usually do not occur during the wet season when LAI is high in semi-arid regions. Globally, burned area showed a bimodal sensitivity to grass height and litter depth (Figure S 14 f-g). In summary, the satellite-derived sensitivities demonstrate a strong global dependence

of burned area dynamics on vegetation type and coverage, litter fuels, pre-season plant productivity and fuel accumulation. FireMIP models reproduced the general increase of burned area with increasing herbaceous vegetation (Figure 4 e, Figure 5). However, regional sensitivities to herbaceous cover differed among models (Figure S 25). The satellite-derived increase of burned area with precedent 6-monthly GPP was reproduced by LPJ_SPITF, ORCHIDEE, JSBACH, and JULES (r > 0.6) while LPJG_SIMF had a reverse relationship (Figure 4 f). However, the FireMIP models underestimated the regional sensitivity to

precedent 6-monthly GPP especially in most fire-prone semi-arid regions such as African savannahs, Australia, the Mediterranean, and temperate steppes (Figure 6 n-o) but patterns strongly differed among models (Figure S 27).

## 4 Discussion and Conclusions

In summary, fire-enabled DGVMs showed the best correlations with monthly observed burned area in some Savannah regions in Africa and South America. However, models generally underestimated the variance of burned area in most fire-prone semi-

arid ecosystems and over-estimated the variance in temperate regions. By using the RF machine learning algorithm, we were able to diagnose reasons for these differences between data and models: Fire-enabled DGVMs largely reproduced data-derived relationships and sensitivities between burned area and climate variables. However, models showed very different relationships with socioeconomic variables and generally underestimated sensitivities to pre-season plant productivity in all semi-arid ecosystems. As a consequence, these results point towards fuel properties and fuel dynamics, and human-fire interactions as

components of fire-enabled DGVMs that should be in the focus of future model development. In the following, we will discuss methodological aspects of our applied pattern-oriented model evaluation approach (4.1), discuss controls on fire in data and models (4.2), and finally provide suggestions on how to improve fire-enabled DGVMs by using current Earth observation datasets (4.3).

### 4.1 Pattern-oriented evaluation of DGVMs using machine learning

Simply speaking, simulations of fire (e.g. burned area) in DGVMs can be wrong because #1 the vegetation model simulates wrong vegetation distributions, plant productivity, and hence fuels, or #2 because the fire module misrepresents the response of fire to weather, humans, or fuel properties. Classical model benchmarking uses, for example, maps of burned area, biomass, and tree cover to quantify the model-data mismatch between these variables (Kelley et al., 2013; Schaphoff et al., 2018). However, classical model benchmarking does not allow to disentangle the individual effects of the vegetation or fire module

on the simulated burned area because errors in the simulated vegetation might be caused by errors in burned area and vice versa. Because we use the same climate forcing, and vegetation state variables derived from each model in our machine



learning approach, we are able to evaluate the response of fire models independent from their underlying DGVMs. This allows us to derive (as partial dependencies or individual conditional expectations) and evaluate the relationships between predictors and response for each fire module separately. Hence we are able to attribute deficiencies in fire-enabled DGVMs to human- and productivity-influences on fire. Previously, a similar approach used also a tree-based machine learning algorithm to

evaluate drivers of soil carbon stocks in observational databases and in Earth System Models (Hashimoto et al., 2017). Unlike classical model benchmarking, such pattern-oriented model evaluation approaches help to diagnose the reasons for model-data mismatches.

The core of our pattern-oriented model evaluation is the application of a machine learning algorithm to learn emergent relationships from data or models. We used the random forest algorithm because this algorithm has been previously used to

identify drivers of burned area (Aldersley et al., 2011; Archibald et al., 2009) and does not require any assumptions about the shape of relationships and the interactions between predictor variables unlike some other algorithms such as generalized additive/linear models (Bistinas et al., 2014; Forkel et al., 2017; Krawchuk et al., 2009). Other flexible algorithms such as maximum entropy have been used as well in empirical fire modelling (Moritz et al., 2012; Parisien et al., 2016) with very similar prediction performance and importance of variables compared to random forest (Arpaci et al., 2014). In addition, the

emergent relationships between predictors and burned area that we identified here show the same directions like the relationships that have been found in a previous study based on generalized linear models (Bistinas et al., 2014). These findings suggest that the choice of the machine learning algorithm only marginally affects the direction and overall shape of the derived relationships.

### 4.2 Controls on burned area

Following previous studies, we found that climate is the primary control of global burned area which affects fire directly through fire weather and fuel moisture conditions, and indirectly through ecosystem productivity, vegetation type, and fuel loads (Archibald et al., 2013; Krawchuk and Moritz, 2011). From the climate variables, the maximum temperature was the dominant predictor globally and especially in northern temperate and boreal ecosystems. Fire-enabled DGVMs generally reproduced the relationships with maximum temperature but on average overestimated the sensitivity in grassland and

Savannah ecosystems. Relationships and sensitivities with the number of wet days showed larger differences among models and in comparison to satellite-derived relationships, suggesting that climate effects on fuel moisture need to be improved in fire-enabled DGVMs.

As an indirect climate effect, we found that previous season plant productivity was among the most important predictor variables globally and was the dominant predictor with the strongest sensitivity to burned area in semi-arid savannah regions.

It has been long recognized that the occurrence and development of fires is affected by the production and accumulation of fuels (Krawchuk and Moritz, 2011; Pausas and Ribeiro, 2013). Plant productivity in fire-prone semi-arid ecosystems has a high year-to-year variability (Ahlström et al., 2015). Our results demonstrate that the inter-annual variability in productivity and hence fuel accumulation is an important driver for the variability in burned area. Most fire-enabled DGVMs poorly



captured the importance, relationship, and sensitivity of previous-season plant productivity on burned area. This may be a reason why they underestimate observed variability in burned area and it might be one reason why they misrepresent trends in fire occurrence in Africa and globally (Andela et al., 2017).

While climate and fuel controls when and where fires can burn, humans are on the one hand responsible for the majority of fire ignitions and on the other hand suppress fire. We found a strong decline of burned area with increasing population density between 0 and 20 person km$^{-2}$ which confirms previous findings (Bistinas et al., 2014; Knorr et al., 2014). Human effects on fire emerge from various activities such as from traditional land use practices (shifting cultivation, hunting, grazing, and grassland burning); the use of fires for land clearing or as tool in land conflicts; from prescribed small fires within fire management; and from unintended or illegal ignitions (Archibald, 2016; Bowman et al., 2011; Lauk and Erb, 2009; Marle et al., 2017). The modest performance of random forest in reproducing satellite burned area suggests that we did not capture the complexity of human-fire interactions with the used set of predictor variables. For example, the complex non-monotonic relationship between burned and cropland cover suggests that agricultural land use has diverging effects on fire in different agricultural regions of the world (Figure S 13 c) (Korontzi et al., 2006). However, alternative variables such as cattle density or the night light-based index of socio-economic development were highly correlated with population density or cropland cover at the coarse resolution of our analysis and did therefore not add to prediction performance of random forest. At regional scales, land use or infrastructure-related variables are important predictors for fire (Archibald et al., 2009; Arpaci et al., 2014; Chuvieco and Justice, 2010; Parisien et al., 2010). However, these regional findings also show that the importance of human-related predictors largely differs between regions, which complicates its applicability for global-scale fire modelling. However, random forest largely emulated the simulated burned area from FireMIP models, which suggests that we indeed included the main predictors for the model world. Although some newer global fire models include effects of cropland and pasture management on fires (Rabin et al., 2018), the complexity of human-fire interactions lacks currently a solid and large-scale empirical basis that would allow to derive alternative formulations on human-fire interactions for fire-enabled DGVMs.

### 4.3 Improving vegetation controls on fire in DGVMs

Our results indicate that the links between vegetation productivity, fuel production, and fire need to be improved to better represent the variability of burned area in fire-enabled DGVMs. Fuel production depends on plant productivity, and on the allocation, turnover, and respiration processes of carbon in different fuel types. As a first step for model improvement, fire-enabled DGVMs need to be tested and possibly re-calibrated against observations or observation-based estimates of plant productivity, above-ground biomass, and carbon turnover (Carvalhais et al., 2014; Thurner et al., 2016, 2017). Beyond total above-ground biomass, the evaluation of different fuel types (e.g. biomass in wood, canopy and understory, and litter size classes) is currently hampered by the availability of data. Only a few in-situ measurements of fuel loads exists (van Leeuwen et al., 2014) and global maps of fuel properties are based on spatial extrapolations including various assumptions and uncertainties (Pettinari and Chuvieco, 2016). As an alternative, hybrid data/model-based approaches such as land carbon cycle





data assimilation systems (Bloom et al., 2016) may provide consistent information to benchmark vegetation productivity, turnover, and litter fuel dynamics in DGVMs.

Fire largely depends on the vegetation type (Rogers et al., 2015). Also our results show consistent land cover-specific relationships to burned area in satellite data, but these relationships differed among FireMIP models and in comparison to the satellite-derived relationships (Figure S 17). Vegetation types and associated morphological, biochemical, and structural characteristics of plants affect the flammability and fire tolerance of vegetation (Archibald et al., 2018; Pausas et al., 2017). Although global fire models have PFT-specific parametrisations for flammability (Thonicke et al., 2010), such fire-relevant plant characteristics need to be incorporated in DGVMs (Zylstra et al., 2016). Such efforts need to be complemented by calibrating DGVMs against satellite observations that provide relevant information about the spatial distributions of fuel structure (Pettinari and Chuvieco, 2016; Riaño et al., 2002), fuel moisture content (Yebra et al., 2013, 2018), fire ignitions and spread (Laurent et al., 2018), fuel consumption (Andela et al., 2016), and fire radiative energy (Kaiser et al., 2012). In summary, besides human-fire interactions, we identified vegetation effects on fire as a main deficiency of fire-enabled dynamic global vegetation models in simulating temporal dynamics of burned area.

**Data availability**

Data is available from the references as indicated in Table A1.

**Code availability**

This analysis is based on R (version 3.3.2) by using the packages randomForest (version 4.6-12) and ICEbox (version 1.1.2). R and the packages are available from the Comprehensive R Archive Network (CRAN, https://cran.r-project.org/).




**Appendix**

**Table A 1: Overview of predictor variables, used datasets, and their use in random forest experiments.**

| Variable | Description | Data source | Variable selection | Use of variables and data sources in random forest (RF) experiments | | |
|---|---|---|---|---|---|---|
| | | | | **RF.Satellite.full** RF with full selected set of observational variables | **RF.Satellite.fm** RF with same variables that are available from FireMIP models | **RF.FireMIP.fm** RF using forcing and outputs from each FireMIP model |
| | Time scale of variables: **(C)** = constant value or multi-year average from model output **(A)** = annual time series **(M)** = monthly time series | | **Correlations in Figure S 1** | | | |
| PopDens | Population density (A) | HYDE V3.1 (Klein Goldewijk et al., 2011) | HYDE | HYDE | HYDE | HYDE |
| GDP | Gross domestic product (C) | W18 (World Bank, 2018) | W18 | W18 | -- | -- |
| NLDI | Night-light development index (C) | E12 (Elvidge et al., 2012) | E12 | -- | -- | -- |
| CattleDens | Cattle density (C) | WR07 (Wint and Robinson, 2007) | WR07 | -- | -- | -- |
| TMAX | Mean of daily maximum temperature (M) | CRUNCEP V5 (Wei et al., 2014) | CRUNCEP | CRUNCEP | CRUNCEP | CRUNCEP |
| TMIN | Mean of daily minimum temperature (M) | | CRUNCEP | -- | -- | -- |
| DTR | Mean of daily diurnal temperature range (M) | | CRUNCEP | CRUNCEP | CRUNCEP | CRUNCEP |
| P | Total precipitation (M) | | CRUNCEP | -- | CRUNCEP | CRUNCEP |
| WET | Number of wet days per month (M) | | CRUNCEP | CRUNCEP | CRUNCEP | CRUNCEP |
| WSPEED | Monthly 90%-ile of daily wind speed (M) | | CRUNCEP | CRUNCEP | CRUNCEP | CRUNCEP |
| Tree.NE | Needle-leaved evergreen trees (A) | CCI: ESA CCI land cover V2.0.7 (ESA CCI-LC, 2017; Li et al., 2018) - OR – FM: Coverage of plant functional types from each FireMIP model | CCI | CCI | -- | FM |
| Tree.ND | Needle-leaved deciduous trees (A) | | CCI | CCI | CCI | FM |
| Tree.BE | Broadleaved evergreen trees (A) | | CCI | CCI | -- | FM |
| Tree.BD | Broadleaved deciduous trees (A) | | CCI | CCI | -- | FM |
| Shrub.BD | Broadleaved deciduous shrubs (A) | | CCI | CCI | -- | -- |
| Shrub.BE | Broadleaved evergreen shrubs (A) | | CCI | CCI | -- | -- |
| Shrub.NE | Needle-leaved evergreen shrubs (A) | | CCI | CCI | -- | -- |
| Herb | Herbaceous vegetation (A) | | CCI | CCI | CCI | FM |
| Crop | Croplands (A) | | CCI | CCI | CCI | FM |
| TreeS.BE | = Tree.BE + Shrub.BE (A) | | -- | -- | CCI | -- |



| | | | | | | |
|---|---|---|---|---|---|---|
| TreeS.BD | = Tree.BD + Shrub.BD (A) | | -- | -- | CCI | -- |
| TreeS.NE | = Tree.NE + Shrub.NE (A) | | -- | -- | CCI | -- |
| GPP.orig | Monthly gross primary production (M) | FLUXCOM: Upscaled GPP based on CRUNCEP climate and satellite data (Tramontana et al., 2016) <br> - OR - <br> FM: simulated GPP from each FireMIP model | FLUXCOM | FLUXCOM | FLUXCOM | FM |
| GPP.pre3mon | Average GPP over the 3 precedent months (M) | | FLUXCOM | -- | FLUXCOM | FM |
| GPP.pre6mon | Average GPP over the 6 precedent months (M) | | FLUXCOM | FLUXCOM | FLUXCOM | FM |
| LAI.orig | Monthly LAI (M) | MODIS | MODIS | MODIS | -- | -- |
| LAI.pre3mon | Average LAI over the 3 precedent months (M) | MODIS | MODIS | -- | -- | -- |
| LAI.pre6mon | AveragLAI over the 6 precedent months (M) | MODIS | MODIS | MODIS | -- | -- |
| cVeg | Total vegetation carbon (C) | C14: Global biomass map (Carvalhais et al., 2014) <br> - OR - <br> FM: simulated cVeg from each FireMIP model | C14 | C14 | C14 | FM |
| CanHeight | Canopy height (C) | S11: Satellite-derived canopy height (Simard et al., 2011) | S11 | -- | -- | -- |
| G_height | Grass height (C) | PC16: Global fuelbed database (Pettinari and Chuvieco, 2016) | PC16 | PC16 | -- | -- |
| L_depth | Litter depth (C) | | PC16 | PC16 | -- | -- |
| W_1h | Woody fuel of the 1 h size class (C) | | PC16 | PC16 | -- | -- |
| W_10h | Woody fuel of the 10 h size class (C) | | PC16 | -- | -- | -- |

**Author contribution**

M. Forkel, NA, SPH, GL and MvM designed the analysis. M. Forkel, SPH, WD and AA defined the overall structure and scope of the manuscript. M. Forkel developed the analysis code and performed the computations. M. Forkel wrote the manuscript with inputs from NA and SPH. GL, M. Forrest, SH, FL, JM, SS and CY made FireMIP model simulations and commented on the study design. EC and AH contributed and processed burned area datasets and related quality layers. WD contributed with predictor datasets. All co-authors commented on the manuscript.

**Acknowledgements**

We thank the institutions, initiatives, and researchers listed in Table A 1 for providing datasets. We thank Stephane Mangeon for providing JULES-INFERNO model results to FireMIP. M. Forkel was supported by a Living Planet Fellowship from the European Space Agency, M. Forkel and W. Dorigo from the TU Wien Wissenschaftspreis 2015, a personal science award to





W. Dorigo. N. Andela received funding from the Gordon and Betty Moore Foundation (grant GBMF3269). The MERIS and

MODIS Fire_cci datasets were generated under the ESA Climate Change Initiative.

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
