# Peer review of "Emergent relationships on burned area in global satellite observations and fire-enabled vegetation models"

_Biogeosciences, 2018_

## Referee Comment (RC1) · Anonymous Referee #1 · 18 Nov 2018

**1   Review bg-2018-427**

**1.1   Emergent relationships on burned area in global satellite observations and fire-enabled vegetation models**

This paper applies a novel and apparently useful methodology, of comparing the performance machine learning (RandomForest) models to fire-enabled DGVMs paramaterised with equivalent driving variables. I feel this paper is a valuable contribution, given the importance of improving fire-enabled DGVMs in order to properly understand

future global-scale shifts in vegetation, climate and fire regime. The paper's methods for assessing variable dependence and importance on a spatial basis are novel and useful. The paper presents a number of useful findings, such as identifying that while DGVM perform comparably to machine learning methods in many area, they tend to poorly capture previous-season productivity drivers of fire - this productivity can be highly variable and a very important driver of fire in arid areas and tropical savannas. As is typical, quantifying the human drivers of ignition patterns through proxies such as population density, GDP etc. proved problematic. This study provides a useful resource for improving global fire and vegetation modelling.

**1.2   Specific Comments**

- The authors used direct rainfall quantities as a predictor variable in the model - both DGVMs and many fire behaviour models and simulation use a measure of soil moisture (as a proxy for fuel moisture) that takes into account rainfall input, and evaporation over time. Soil moisture content is also a variable available in global climate model output. Why did the authors not include soil moisture content as a predictor variable in the random forest modelling, in addition to precipitation, as it may provide a more physically relevant correlate of fire activity?

- What defines the white, presumably "missing data" cells in maps, eg. Figure 2? It is notable that no data appears availabe for south-eastern and south-western Australia, which are both fire-prone areas with comparable climates to eg. South Africa, western United States, mediterranean. I see tht there is a brief comment in the caption for figure 6 explaining there was "missing data" in the vegetation carbon dataset for Australia and New Zealand, but I feel this needs more explanation given the importance of fire in this area. What would need to be done to include this area - is there another potential data source that could be used as a replacement?

[Figure]

**1.3 Technical corrections**

- P 2 L 3: Allowed *us* to diagnose...

- P 3 L 16: potential drivers *of* fire...

- P 9 L 25: same range *as* the correlation...

- Supplementary figure S16 (a) - The labels for the different model lines are poorly justified and are clipped by the edge of the figure. I'm also not sure why these labels need to be repeated in panel (f). Figure 17 has similar issues - the choice of which panel(s) to include this legend in seems arbitrary.

---

## Referee Comment (RC2) · Anonymous Referee #2 · 26 Nov 2018

The paper is a welcome development mostly because it largely highlights the inadequacies of DGVMs, but also recommends necessary changes. The section with recommendations for specific changes in DGVMs, backed up with the current findings, make this paper novel and very useful for the modeling and the fire data community. However, the analysis is mostly confirming prior findings on drivers of burned area at global scale. One additional take home message, perhaps the most important after reading the manuscript, was that "besides human-fire interactions, we identified vegetation effects on fire as a main deficiency of fire-enabled dynamic global vegetation models in simulating temporal dynamics of burned area."

[Figure]

Specific Comments

- In the abstract (ln 27) you mention "Recent climate changes increases fire-prone weather conditions and likely affects fire occurrence". That is not happening everywhere in the globe. Please make clear if that's the case globally or regionally, or in certain latitudes ect.

- Ln 10: Regionally (Randerson et al., 2006), but also globally (Lopez-Saldaña et al., 2015 : https://www.biogeosciences.net/12/557/2015/)

- Ln 14: you can add Veraverbeke et al., 2016 (https://www.nature.com/articles/nclimate3329), for changes in lightning ignitions.

- Pg 3 – ln 8 and ln 10: "...drivers of fire activity". There has always been confusion on what fire activity, or fire incidence really represents. Do you study drivers of fire activity, or drivers of burned area?

- Pg 5 – ln 20: I Suspect there might be some uncertainty from aggregating different model resolutions. Could please comment on that? For example, type of aggregation (nearest neighbor ect), latitude/area correction ect.

- Pg 7 - ln 15: Why a Spearman correlation of >0.25 is considered as good? Because >0.25 agreement in spatial patterns is acceptable? Most of the BA is happening in Africa anyway and this is an area that burns frequently and therefore most products and models mostly agree. But that also means that more sparse events (boreal fires) show a very low agreement. So, is that what you mean? That given the fact that the very general patterns are described, a >0.25 correlation is a good correlation?

- Pg 7 – ln 16: You do not describe how do you treat your data priory to the analysis. Data like population density are extremely skewed. Did you apply any transformation to the data, checked for outliers, false alarms ect?

- Pg 8 – ln 26: The GDP data, especially in areas of high fire activity (Africa) are based in country averages. In this case it is weighted by the pop. density. That means that it

is eventually following the variability of the pop. density. Did you see this effect, and if yes, did you take it into account?

- Figure 3: I feel it is a bit misleading to say that vast areas in temperate and boreal regions in north hemisphere are temperature driven. Essentially, this means fuel moisture variability and the monthly effect of moisture conditions rather than extreme temperatures (of DTR) that might last only few days. You mentions this somehow in pg15-ln5 and pg19-ln20 on, but please make this a bit more clear in the discussion if it is the case or not..

- The take home message (pg21-ln12), perhaps the most important after reading the manuscript, was that ". . . we identified vegetation effects on fire as a main deficiency of fire-enabled dynamic global vegetation models in simulating temporal dynamics of burned area." It would be great if that would be more highlighted in the discussion.

---

## Author Comment (AC1) · 4 Dec 2018

We thank the referee for the positive review. In the following we cite comments by the referee and provide our responses in normal font.

[Referee 1: "The authors used direct rainfall quantities as a predictor variable in the model - both DGVMs and many fire behaviour models and simulation use a measure of soil moisture (as a proxy for fuel moisture) that takes into account rainfall in- put, and evaporation over time. Soil moisture content is also a variable available in global climate model output. Why did the authors not include soil moisture content as a predictor variable in the random forest modelling, in addition to precipitation, as it may

provide a more physically relevant correlate of fire activity?"]

Initially, we indeed considered to use soil moisture as predictor for burned area. However, the FireMIP models only provided simulations of soil moisture as the total soil moisture content (kg m-2) integrated over the entire soil depth. Each model uses a different definition for soil depth and also the soil hydrology schemes differ. Hence soil moisture simulations from the different models are not directly comparable. The relative soil moisture of the upper soil layer will be a new output from FireMIP models in a next release of model results.

We also rely in our analysis for each predictor variable on an observational dataset or data-driven estimate. For soil moisture, we initially used the ESA CCI soil moisture dataset which is an estimate of volumetric surface (upper < 5 cm) soil moisture ($m^3/m^3$) from active and passive microwave satellites (Dorigo et al., 2017). We have been shown that the ESA CCI soil moisture dataset can be used as a predictor for burned area at the global scale (Forkel et al., 2017). However, we also found that the number of wet days resulted in better performances at the global scale. We did not further use the ESA CCI soil moisture dataset in this study because the volumetric surface soil moisture (ESA CCI) is only to a limited extent comparable with the column-integrated total soil moisture content from the FireMIP models. In summary, we did not include soil moisture as predictor because of the limited comparability of observational and modelled soil moisture datasets.

[Referee 1: "What defines the white, presumably "missing data" cells in maps, eg. Figure 2? It is notable that no data appears availabe for south-eastern and south-western Australia, which are both fire-prone areas with comparable climates to eg. South Africa, western United States, mediterranean. I see tht there is a brief comment in the caption for figure 6 explaining there was "missing data" in the vegetation carbon dataset for Australia and New Zealand, but I feel this needs more expla- nation given the importance of fire in this area. What would need to be done to include this area - is there another potential data source that could be used as a replacement?"]

We used a map of forest and herbaceous vegetation carbon as predictor for burned area. This map (Carvalhais et al., 2014) is based on the map of tropical aboveground forest biomass by Saatchi et al. (2011). The Saatchi et al. map does not cover southern Australia and New Zealand and hence these regions appear as "missing data" in the random forest-based results. Alternative maps of tropical forest biomass do not include Southern Australia and New Zealand (Avitabile et al., 2016; Baccini et al., 2012) and global maps of forest biomass do not account for herbaceous biomass (e.g. http://globbiomass.org/products/global-mapping/). Regional maps of vegetation carbon or forest biomass might be available for Australia and could be potentially mosaicked with the used global map. However, such a mosaicking will likely introduce artefacts (e.g. edges between both biomass maps) that might propagate in maps of controls on burned area and associated data-model comparisons. Because of these reasons, we did not try to create or integrate an additional vegetation carbon map that covers the missing regions. Global maps of vegetation carbon will hopefully become available with recently launched (e.g. ICESat-2) and future satellite missions (e.g. BIOMASS). We now provide the following explanation in chapter 2.5:

"The vegetation biomass dataset does not cover southern Australia and New Zealand. Although fire is common in these regions, we did not fill the global vegetation biomass map with a regional map to avoid potential artefacts in the derived sensitivities that would likely result from merging different biomass maps."

In addition, we added the following sentence in the captions of Figures 2, 3 and 6:

"Regions with missing data (white) are either without vegetation cover (e.g. deserts, ice sheets), had not burned area (e.g. parts of the Amazon and tundra), or were not covered by the used vegetation carbon map (i.e. regions in southern Australia and New Zealand)."

We changed the proposed Technical Corrections.

References

[Figure]

Avitabile, V., Herold, M., Heuvelink, G. B. M., Lewis, S. L., Phillips, O. L., Asner, G. P., Armston, J., Ashton, P. S., Banin, L., Bayol, N., Berry, N. J., Boeckx, P., Jong, D., J, B. H., DeVries, B., Girardin, C. A. J., Kearsley, E., Lindsell, J. A., Lopez‐-Gonzalez, G., Lucas, R., Malhi, Y., Morel, A., Mitchard, E. T. A., Nagy, L., Qie, L., Quinones, M. J., Ryan, C. M., Ferry, S. J. W., Sunderland, T., Laurin, G. V., Gatti, R. C., Valentini, R., Verbeeck, H., Wijaya, A. and Willcock, S.: An integrated pan‐tropical biomass map using multiple reference datasets, Glob. Change Biol., 22(4), 1406–1420, doi:10.1111/gcb.13139, 2016.

Baccini, A., Goetz, S. J., Walker, W. S., Laporte, N. T., Sun, M., Sulla-Menashe, D., Hackler, J., Beck, P. S. A., Dubayah, R., Friedl, M. A., Samanta, S. and Houghton, R. A.: Estimated carbon dioxide emissions from tropical deforestation improved by carbon-density maps, Nat. Clim. Change, 2(3), 182–185, doi:10.1038/nclimate1354, 2012. Carvalhais, N., Forkel, M., Khomik, M., Bellarby, J., Jung, M., Migliavacca, M., ÎlJu, M., Saatchi, S., Santoro, M., Thurner, M., Weber, U., Ahrens, B., Beer, C., Cescatti, A., Randerson, J. T. and Reichstein, M.: Global covariation of carbon turnover times with climate in terrestrial ecosystems, Nature, 514(7521), 213–217, doi:10.1038/nature13731, 2014.

Dorigo, W., Wagner, W., Albergel, C., Albrecht, F., Balsamo, G., Brocca, L., Chung, D., Ertl, M., Forkel, M., Gruber, A., Haas, E., Hamer, P. D., Hirschi, M., Ikonen, J., de Jeu, R., Kidd, R., Lahoz, W., Liu, Y. Y., Miralles, D., Mistelbauer, T., Nicolai-Shaw, N., Parinussa, R., Pratola, C., Reimer, C., van der Schalie, R., Seneviratne, S. I., Smolander, T. and Lecomte, P.: ESA CCI Soil Moisture for improved Earth system understanding: State-of-the art and future directions, Remote Sens. Environ., doi:10.1016/j.rse.2017.07.001, 2017.

Forkel, M., Dorigo, W., Lasslop, G., Teubner, I., Chuvieco, E. and Thonicke, K.: A data-driven approach to identify controls on global fire activity from satellite and climate observations (SOFIA V1), Geosci Model Dev, 10(12), 4443–4476, doi:10.5194/gmd-10-4443-2017, 2017.

Saatchi, S. S., Harris, N. L., Brown, S., Lefsky, M., Mitchard, E. T. A., Salas, W., Zutta, B. R., Buermann, W., Lewis, S. L., Hagen, S., Petrova, S., White, L., Silman, M. and Morel, A.: Benchmark map of forest carbon stocks in tropical regions across three continents, Proc. Natl. Acad. Sci., 108(24), 9899–9904, doi:10.1073/pnas.1019576108, 2011.

---

## Author Comment (AC2) · 4 Dec 2018

We thank the referee for the positive review. In the following we cite comments by the referee and provide our responses in normal font.

[Referee 2: "In the abstract (ln 27) you mention "Recent climate changes increases fire-prone weather conditions and likely affects fire occurrence". That is not happening every- where in the globe. Please make clear if that's the case globally or regionally, or in certain latitudes ect."]

The referee it right. Increases in fire-prone weather conditions (estimated as changes

fire weather season length, 1979-2013) occur in 25% of the global vegetated area, especially in eastern and southern Europe, the Mediterranean, Mongolia, and Eastern China, central and eastern Africa, Alaska, the USA and Mexica, and in central and eastern Brazil and Argentina (Jolly et al., 2015). Many regions show no significant changes in fire weather season length and some regions (10.7%) show declines in fire weather season length (e.g. western Africa, parts of Central Asia) (Jolly et al., 2015). We changed the sentence:

"Recent climate changes increases fire-prone weather conditions in many regions and likely affects fire occurrence"

[Referee 2: " Ln 10: Regionally (Randerson et al., 2006), but also globally (Lopez-Saldaña et al., 2015 : https://www.biogeosciences.net/12/557/2015/)]

Thank you for this interesting reference. We changed the sentence: "Fire affects global and regional and climate directly through changing surface albedo (López-Saldaña et al., 2015; Randerson et al., 2006)"

[Referee 2: " Ln 14: you can add Veraverbeke et al., 2016 (https://www.nature.com/articles/nclimate3329), for changes in lightning ignitions.]

We added the reference.

[Referee 2: "Pg 3 – ln 8 and ln 10: ". . .drivers of fire activity". There has always been confusion on what fire activity, or fire incidence really represents. Do you study drivers of fire activity, or drivers of burned area?]

We replaced "fire activity" with "burned area".

[Referee 2: "Pg 5 – ln 20: I Suspect there might be some uncertainty from aggregating different model resolutions. Could please comment on that? For example, type of aggregation (nearest neighbor ect), latitude/area correction ect."]

We added the following sentences: "Aggregation was done by averaging the fractional

burned area from all high-resolution grid cells that belong to the same coarse-resolution grid cell. Nearest neighbour resampling was done if less than two high-resolution grid cells were within one coarse-resolution grid cell."

[Referee 2: "Pg 7 - ln 15: Why a Spearman correlation of >0.25 is considered as good? Because >0.25 agreement in spatial patterns is acceptable? Most of the BA is happening in Africa anyway and this is an area that burns frequently and therefore most products and models mostly agree. But that also means that more sparse events (boreal fires) show a very low agreement. So, is that what you mean? That given the fact that the very general patterns are described, a >0.25 correlation is a good correlation?"]

The Spearman correlation between simulated and observed burned area was computed per grid cell (not across the spatial patterns). It is easier for the models to reach a higher correlation in regions with frequent fires that have a clear seasonality of burned area such as African Savannahs than in regions with infrequent burning such as boreal forests. Hence we used such a low threshold of the correlation coefficient to define a "good" correlation because this shows at least that a model has a positive relationship with observed burned area. For regional studies with frequent burning a higher threshold of the correlation would be more appropriate. We changed the sentence to: "good agreement for an individual grid cell was defined based on a positive and non-random relationship (i.e. $Cor \geq 0.25$) and a comparable variance ($-0.75 \leq FV \leq 0.75$) between simulated and observed burned area."

[Referee 2: "Pg 7 – ln 16: You do not describe how do you treat your data priory to the analysis. Data like population density are extremely skewed. Did you apply any transformation to the data, checked for outliers, false alarms ect?"]

We did not apply any data transformations such as normalization or scaling because the random forest algorithm works also with non-normal distributed data. We applied quality checks to the burned area data as described in chapter 2.2.

[Referee 2: " Pg 8 – ln 26: The GDP data, especially in areas of high fire activity (Africa) are based in country averages. In this case it is weighted by the pop. density. That means that it is eventually following the variability of the pop. density. Did you see this effect, and if yes, did you take it into account?"]

The GDP data was normalized by population density (GDP per capita). Hence any population density-related variability that was included in the original GDP dataset should be taken into account. We added in Table A1 that GDP is per capita.

[Referee 2: "Figure 3: I feel it is a bit misleading to say that vast areas in temperate and bo- real regions in north hemisphere are temperature driven. Essentially, this means fuel moisture variability and the monthly effect of moisture conditions rather than extreme temperatures (of DTR) that might last only few days. You mentions this somehow in pg15-ln5 and pg19-ln20 on, but please make this a bit more clear in the discussion if it is the case or not.. "]

We did not write that fire "in temperate and boreal regions (. . .) are temperature driven". This reading of our manuscript likely emerged from the maps in Figure 3 and the associated text where we only describe the most important predictor in the random forest model but do not provide information on the ranking of the other predictors. We did the following changes to avoid such an interpretation:

- In the caption of Figure 3, we added: "Please note that the predicted burned area in random forest (and in reality) emerges from multiple predictors and that the second-most important predictor (not shown in the maps) might have similar importance."

- We changed the sentence at page 19, line 20 to "Fire results from an interplay of several meteorological variables, thereby maximum temperature was an important predictor globally and especially in northern temperate and boreal ecosystems."

[Referee 2: "The take home message (pg21-ln12), perhaps the most important after reading the manuscript, was that ". . . we identified vegetation effects on fire as a

main deficiency of fire-enabled dynamic global vegetation models in simulating temporal dynamics of burned area." It would be great if that would be more highlighted in the discussion."]

To better highlight this take-home message, we made the following changes:

- We added the following sentence in the beginning of chapter 4.3. "Our results demonstrate that the role of vegetation on fire needs to be better represented in fire-enabled DGVMs to accurately simulate the variability of burned area."

- We changed the last sentence of the abstract: "Hence our pattern-oriented model evaluation approach allowed us to diagnose that vegetation effects on fire are a main deficiency of fire-enabled dynamic global vegetation models to accurately simulate the role of fire under global environmental change."

References

Jolly, W. M., Cochrane, M. A., Freeborn, P. H., Holden, Z. A., Brown, T. J., Williamson, G. J. and Bowman, D. M. J. S.: Climate-induced variations in global wildfire danger from 1979 to 2013, Nat. Commun., 6, 7537, doi:10.1038/ncomms8537, 2015.

López-Saldaña, G., Bistinas, I. and Pereira, J. M. C.: Global analysis of radiative forcing from fire-induced shortwave albedo change, Biogeosciences, 12(2), 557–565, doi:https://doi.org/10.5194/bg-12-557-2015, 2015.

Randerson, J. T., Liu, H., Flanner, M. G., Chambers, S. D., Jin, Y., Hess, P. G., Pfister, G., Mack, M. C., Treseder, K. K., Welp, L. R., Chapin Iii, F. S., Harden, J. W., Goulden, M. L., Lyons, E., Neff, J. C., Schuur, E. A. G. and Zender, C. S.: The Impact of Boreal Forest Fire on Climate Warming, Science, 314(5802), 1130–1132, 2006.

---

## Author Response (AR1)

**Emergent relationships on burned area in global satellite observations and fire-enabled vegetation models**

Matthias Forkel[1], Niels Andela[2], Sandy P. Harrison[3], Gitta Lasslop[4], Margreet van Marle[5], Emilio Chuvieco[6], Wouter Dorigo[1], Matthew Forrest[4], Stijn Hantson[7], Angelika Heil[8], Fang Li[9], Joe Melton[10], Stephen Sitch[11], Chao Yue[12], and Almut Arneth[13]

**Response to reviews**

We thank both referees for their positive feedback. Our responses to the reviews RC1 and RC2 are included in the author comments AC1 and AC2, respectively:

**https://www.biogeosciences-discuss.net/bg-2018-427/#discussion**

**List of changes**

Among other small changes, we made the following changes (page and line numbers refer to the new version of the manuscript):

- P 2, L 3-5: "Hence our pattern-oriented model evaluation approach allowed us to diagnose that vegetation effects on fire are a main deficiency of fire-enabled dynamic global vegetation models to accurately simulate the role of fire under global environmental change."

- P 2, L 9-10: "Fire affects global and regional climate directly through changing surface albedo (López-Saldaña et al., 2015; Randerson et al., 2006)"

- P 2, L 14-15: "Climate influences several aspects of the fire regime, including the seasonal timing of lightning ignitions (Veraverbeke et al., 2017), temperature and moisture controls on fuel drying, and wind-driven fire spread (Jolly et al., 2015)."

- P 3, L 8: We replaced "fire activity" with "burned area".

- P 5, L 22-22: "Aggregation was done by averaging the fractional burned area from all high-resolution grid cells that belong to the same coarse-resolution grid cell. Nearest neighbour resampling was done if less than two high-resolution grid cells were within one coarse-resolution grid cell."

- P 7, L 13-16: "As a single global agreement metric, we computed the percentage of the land area that shows a "good" agreement from the spatial patterns of Spearman correlation Cor and FV, where good agreement for an individual grid cell was defined based on a positive and non-random relationship (i.e. Cor $\geq$ 0.25) and a comparable variance (-0.75 $\leq$ FV $\leq$ 0.75) between simulated and observed burned area."

- P 8, L 22-24: "The vegetation biomass dataset does not cover southern Australia and New Zealand. Although fire is common in these regions, we did not fill the global vegetation biomass map with a regional map to avoid potential artefacts in the derived sensitivities that would likely result from merging different biomass maps."

- P 11, L3-4 and P 13, L 11-13 and P 17, L 3-5: "Regions with missing data (white) are either without vegetation cover (e.g. deserts, ice sheets), had no burned area (e.g. parts of the Amazon and tundra), or were not covered by the used vegetation carbon map (i.e. regions in southern Australia and New Zealand)."

- P 13, L 6-8: "Please note that the predicted burned area in random forest (and in reality) emerges from multiple predictors and that the second-most important predictor (not shown in the maps) might have similar importance."

- P 19, L 23-24: "Fire results from an interplay of several meteorological variables, thereby maximum temperature was an important predictor globally and especially in northern temperate and boreal ecosystems."

[revised manuscript text omitted]